

# Quenching Factor estimation of Na recoils in NaI(Tl) crystals using a low-energy pulsed neutron beam measurement

M. R. Bharadwaj[1*], G. Angloher[1], P. S. Barbeau[2,3], I. Dafinei[4],
N. Di Marco[5,6], L. Einfalt[7,8], F. Ferroni[4,5], S. Fichtinger[7], A. Filipponi[6,9], T. Frank[1],
M. Friedl[7], A. Fuss[7,8], Z. Ge[10], S. Hedges[2,3], M. Heikinheimo[11], K. Huitu[11],
M. Kellermann[1], R. Maji[7,8], M. Mancuso[1], L. Pagnanini[5,6], F. Petricca[1], S. Pirro[6],
F. Pröbst[1], G. Profeta[6,9], A. Puiu[5,6], F. Reindl[7,8], K. Schäffner[1], J. Schieck[7,8],
D. Schmiedmayer[7,8], C. Schwertner[7,8], M. Stahlberg[1], A. Stendahl[11], M. Stukel[5],
F. Wagner[7], S. Yue[10], V. Zema[1] and Y. Zhu[10]

⋆ mukund@mpp.mpg.de

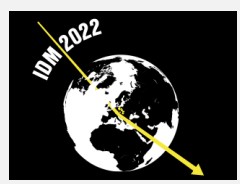

## Abstract

NaI(Tl) based scintillation detectors have become a staple in the field of direct dark matter searches, with the DAMA-LIBRA experiment being the standout for its reported observation of an annually modulating WIMP-like signal which is in direct contrast with other results. In order to accurately calibrate the energies of WIMP-induced nuclear recoil signals, precise measurements of the quenching factor of the NaI crystals are essential for each of these experiments, as it is well established that electron recoils and nuclear recoils have dissimilar scintillation light yields. In this contribution, we present first preliminary results of an ongoing systematic study that has been carried out by the COSINUS collaboration and Duke University to measure the quenching factor of Na recoils. Five ultra-pure NaI crystals, manufactured by the Shanghai Institute for Ceramics, each of which have varying Tl dopant concentrations, were irradiated with a mono-energetic neutron beam at the Triangle Universities National Laboratory, North Carolina, USA to extract the quenching factor values in the low recoil energies of 1-30keV$_{nr}$.

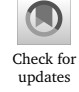

**1** Max-Planck-Institut für Physik, 80805 München - Germany
**2** Department of Physics, Duke University, Durham, NC 27708, USA
**3** Triangle Universities Nuclear Laboratory, Durham, NC 27708, USA
**4** INFN - Sezione di Roma, 00185 Roma - Italy
**5** Gran Sasso Science Institute, 67100 L'Aquila - Italy
**6** INFN - Laboratori Nazionali del Gran Sasso, 67010 Assergi
**7** Institut für Hochenergiephysik der Österreichischen Akademie der Wissenschaften,
1050 Wien - Austria

**8** Atominstitut, Technische Universität Wien, 1020 Wien - Austria
**9** Dipartimento di Scienze Fisiche e Chimiche, Università degli Studi dell'Aquila,
67100 L'Aquila - Italy
**10** SICCAS - Shanghai Institute of Ceramics, Shanghai - P.R.China 200050
**11** Helsinki Institute of Physics, 00014 University of Helsinki - Finland

# 1 Introduction

The DAMA/LIBRA experiment has reported a consistent, annually-periodic modulation signal expected from a dark matter interaction, with latest results reporting a $13.7\sigma$ [1] statistical significance. However, several experiments utilizing different target materials over the years have reported no such observations, casting doubt over the DAMA claim under the standard WIMP scenario. New experiments which utilize the same absorber material: NaI, such as COSINUS (Cryogenic Observatory for SIgnatures seen in Next-generation Underground Searches) [2] and SABRE (Sodium-iodide with Active Background REjection) [3] are currently being setup while the ANAIS-112 (Annual modulation with NaI Scintillators) and COSINE-100 [4,5] are in the data-taking phase to provide a model-independent cross-check of the reported results. With the exception of the COSINUS experiment which has the novel ability to estimate the quenching factor of the crystals in-situ during operation, the signal interpretation and the WIMP parameter space a particular NaI-based experiment covers has a very strong dependence on the nuclear recoil quenching factor. This is because the standard WIMP scattering interactions are assumed to primarily take place via nuclear recoils, while the detectors themselves are calibrated using $\gamma$-sources which interact primarily through energy transfer via electron recoils. Now, it is known that nuclear recoils ($E_{nr}$) in NaI absorbers produce less scintillation light when compared with electron recoils ($E_{ee}$) having an equivalent energy deposition. This ratio of the two, dubbed as the quenching factor, is utilized to appropriately scale the WIMP scattering spectrum. Previous studies [6] discussed how an energy-dependent quenching factor in the 2-6 keV$_{ee}$ could shift the energy window of the modulating signal that DAMA/LIBRA observes from it's assumed 7-20 keV$_{nr}$ to 13-32 keV$_{nr}$. Various measurements carried out over the years have reported Na recoil quenching factors in NaI(Tl) crystals with a significant relative uncertainty, especially in the low energy recoil range- [7–9]. While the method of crystal growth itself can impart a variability of the quenching factor between individual experiments, there is also no estimate on the Tl dopant concentrations of the tested crystals. This adds another layer of uncertainty in understanding and reconciling the results put forth. The preliminary analysis presented in this contribution is the first step of a systematic study to measure the QF as a function of the Tl dopant concentration. For this purpose, 5 highly radio-pure NaI crystals were manufactured by the Shanghai Institute for Ceramics (Shanghai, China), each with differing Tl dopant concentrations to try and extract the quenching factors for each crystal in the 1-30keV$_{nr}$ recoil energy range. The experiment was carried out using a pulsed neutron beam generated at the Neutron calibration facility at Triangle Universities Nuclear Laboratory (TUNL), Duke University (North Carolina, US).

# 2 Experimental setup

TUNL consists of 3 main accelerator facilities, namely: The Tandem Accelerator lab, The High Intensity $\gamma$-ray Source (HIGS) and the Laboratory for Experimental Nuclear Astrophysics (LENA).

Mono-energetic neutrons that are required for quenching factor measurements are produced at the tandem laboratory using an FN tandem Van de Graff accelerator that can deliver a maximum terminal voltage of 10 MV. A pulsed proton beam is created using a Direct Extraction Negative Ion Source (DENIS) to generate negative ions, which are then accelerated via the Van de graff accelerator.

Once the resultant pulsed beam of protons arrives at the quenching factor station, a 1434 nm LiF foil evaporated onto a thin Ta substrate placed at the target location is irradiated by the beam. Resultant mono-energetic neutrons with an energy of ~1300 keV (with a small spread due to proton energy loss in LiF) are produced via the $^7$Li$(p,n)^7$Be reaction. A Beam Pulse Monitor (BPM) was used in order to record the timing information as to when the pulsed proton beam interacted with the LiF target, thus giving the timing information about when the neutrons were produced.

A bi-layer shielding consisting of high density Polyethylene(HDPE) and borated HDPE was placed around the enclosure of the LiF target with a collimated slit to direct the beam towards the NaI crystal. The resultant collimated beam had an angular spread of 2.356°. An additional layer of lead(4 inches thickness) covered the front surface in order to reduce the fraction of secondary gammas produced by the neutron capture of hydrogen that reaches the detector.

## 2.1 Detector setup and data acquisition

The experimental setup for the measurement is as shown in Figure 1. 15 scintillating "backing" detectors (BD) consisting of EJ-309 liquid scintillation cells were deployed for the current run to tag the scattered neutrons off the Na or I nuclei. Each of the BDs was equipped with a lead shielding cap in front of their enclosure during operation to reduce the background gamma trigger rate. Further, an additional backing detector was employed as a time-of-flight detector in order to measure and monitor the spread of the neutron beam energy.

The NaI crystals were located at a distance of 75cm from the LiF target in line with the beam axis. The NaI crystals, manufactured at the Shanghai Institute for Ceramics, China were produced using "Astro-Grade" powder procured from Merck Co. (previously Sigma Aldrich). Inductively coupled plasma mass spectrometry (ICP-MS) measurements performed at LNGS showed contamination at a level of 10 ppb, 0.1 ppb and 0.2 ppb for $^{40}$K, $^{232}$Th and $^{238}$U respectively. Overall, 5 samples were prepared, with their Tl dopant levels varying from 0.1, 0.3, 0.5, 0.7 to 0.9% respectively in the initial powder.

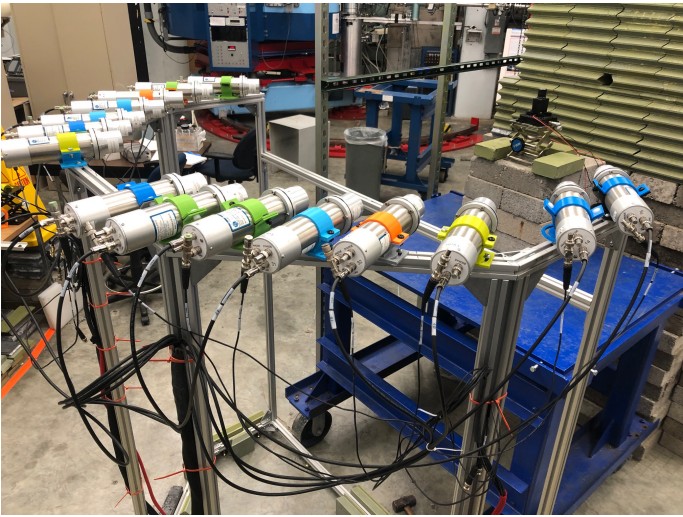

Figure 1: Experimental setup at TUNL

## 3 Data analysis

For data acquisition, a pair of SIS3316 14-bit digitizers which have a sampling rate of 250 MHz was used to collect, acquire and record the data from all the 15 backing detectors (one of which was utilized as the TOF detector), the beam pulse monitor and the NaI PMT respectively whenever a single BD was triggered. In order to accurately identify and filter out only neutron-induced nuclear recoil events in the NaI(Tl) detector, a coincidence trigger between the NaI(Tl) detector and one of the backing detectors was used. For every coincidence event, we first identified which backing detector contributed to the PMT trigger by comparing the pulse onset timing information with the coincidence trigger time (∼350 ns depending on BD position). If the signals from the BD and the PMT associated with the NaI(Tl) detector satisfied certain event selection criteria, the NaI(Tl) signal and the corresponding BD number was saved for further analysis. A finite window integration scheme was initially used to reconstruct the recorded pulses, which was later changed to the adopted charge estimate method as outlined in Ref. [9] which allowed for a much better reconstruction of the low energy pulses.

### 3.1 Identification of neutron-induced nuclear recoils

Utilizing the property of liquid scintillation detectors that they produce a characteristic time distribution of the scintillation light based on the type of interacting particle, we can apply corresponding cuts based on the charge comparison method to select only scattered neutron hits and reject any accidental triggers due to ambient/scattered gammas. The Pulse Shape Discrimination (PSD) was performed using the ratio of the charge sum of the tail to the total charge. The PSD plot with the applied cuts for a given crystal and BD is shown in Fig. 2.

An additional time-of-flight (TOF) cut is also applied in the next step. This is particularly helpful as the TOF of the scattered neutrons from the Na/I nuclei to the BD is almost constant for an incident mono-energetic neutron beam(with a small spread due to a spread in the initial neutron beam energy). Thus, it ensures that we remove any accidental triggers due to neutrons that may have scattered off different parts of the experimental setup and not off the crystal.

### 3.2 Detector calibration and simulation studies

For calibration of the PMT coupled to the individual NaI crystals, a set of $^{133}$Ba, $^{137}$Cs and $^{241}$Am gamma ray sources were used at the beginning and end of each individual run in order

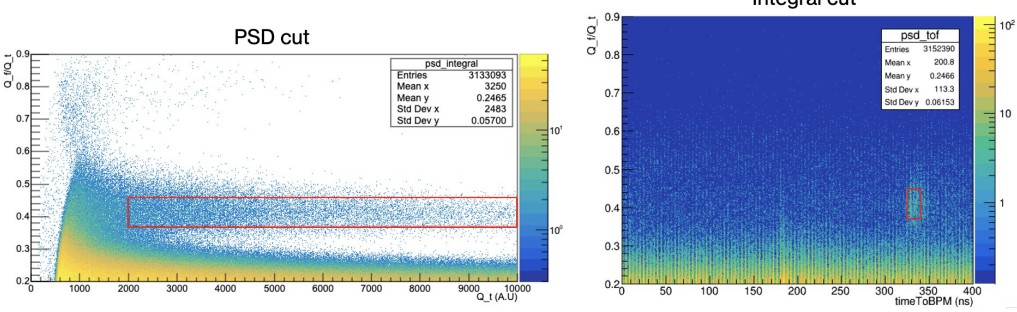

Figure 2: A snapshot of the cuts applied for neutron-only recoils event selection. The PSD cut ensures that only neutron induced events are selected, but this invariably also accepts any secondary scattered neutrons. An additional Integral cut is thus applied which significantly reduces backgrounds from accidental neutrons, gammas and scatters off the experimental setup.

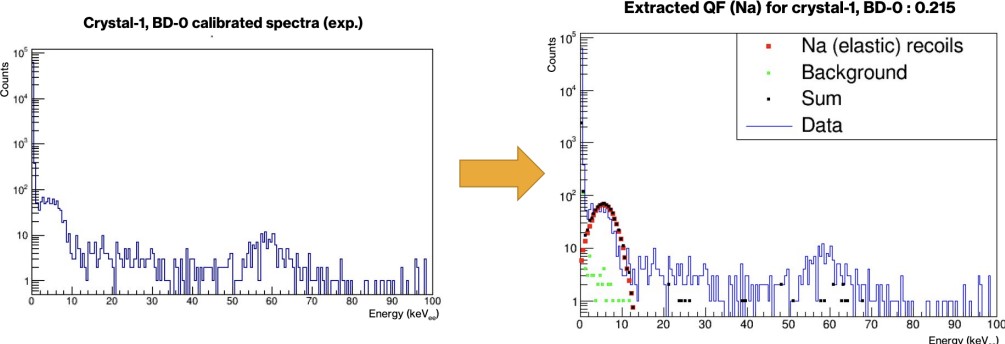

Figure 3: Estimation of quenching factor for elastic nuclear recoils off Na Nuclei for crystal 1 and backing detector 0. The fit was performed using a negative log-likelihood minimization algorithm and yielded a quenching factor of 0.215 for recoils off of Na nuclei.

to set the electron-equivalent energy scale. Initially, in this preliminary analysis, a linear energy calibration function using multiple low energy X-ray peaks from $^{133}$Ba and $^{241}$Am was chosen. The energy of the observed X-ray peaks were cross-checked with GEANT4 simulations of the calibration setup. A more detailed study with different calibration functions will be discussed in a future publication.

Once the cleaned experimental data set was calibrated using the electronic-equivalent energy scale ($E_{ee}$), a GEANT4 simulation of the entire experimental setup incorporating all the various elements such as the backing detectors, the NaI detector housing, the bi-layer collimator and the neutron source was carried out. Using this, a simulated nuclear recoil spectrum representative of the true nuclear energy scale ($E_{nr}$) was generated for each backing detector respectively.

### 3.3  Quenching factor estimation

The Na-recoil quenching factors could be extracted by fitting the experimental cleaned and calibrated recoil spectra ($E_{ee}$) to a gaussian distribution (taking into consideration the background noise distribution) and comparing the mean of the spectrum to the mean of the simulated gaussian-fitted spectra obtained via simulation ($E_{nr}$), with the ratio of the two giving the quenching factor. A negative log-likelihood algorithm was implemented to estimate the best fit. The simulated spectra incorporates the information contained in the inelastic recoil peaks and various effects like scattering from the collimator, influence of the NaI housing material and other related systematics which are already factored in the simulation. An example of the fit process is shown for crystal 1 and recoil events associated with backing detector 0.

## 4  Conclusions and future work

The initial work presented in this study verified the analysis workflow being implemented for the extraction of the quenching factor for Na and I recoils. First preliminary results of the quenching factor for Na recoils in the 10-30 keV$_{nr}$ energy regime yields a value in the range of 0.2 for the different crystals tested assuming a multi-point linear calibration scheme which is in line with the reported values calculated in [10]. A further analysis accounting for all the systematic uncertainties, PMT threshold effects and optimizing the cut parameters and fits is currently in the final stages and a future publication describing the possible influence

of Tl dopant concentration on said results and the impact of different calibration schemes is currently underway.

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
