# Peer review of "Quenching Factor estimation of Na recoils in NaI(Tl) crystals using a low-energy pulsed neutron beam measurement"

_SciPost Physics Proceedings, doi:SciPost Phys. Proc. 12, 028 (2023)_

## Round 1 · Referee Report · Anonymous (Referee 1) · 2022-10-12

Report
The paper, contribution to the Proceedings of the IDM 2022 conference, shows the status of the NaI quenching factors measurements at TUNL for various concentrations of Tl dopant. The paper can be published, once the small remarks in the following will be fulfilled.
-
in the abstract it is written: “… and conclusively rule out the parameter space …” Generally, the experiments are done to confirm or disprove, never only to rule out. Thus I suggest a more fair sentence.
-
Line 44: …potentially mimics… A more polite sentence is required
-
lines 61,62. Actually DAMA papers report three different options for the quenching factors: 1) those independent of the recoil energy; 2) including the channeling effect; 3) using the energy dependence of the Tretyak model. Thus, the sentence must be modified accordingly.
-
lines 66-68. Other levels of uncertainty in the q.f. arise from the other features of the used NaI detectors. The Tl dopant is just one of them. The method of growth of the crystal itself plays a relevant role too.
-
line 88. The range of the neutron energy produced by the facility should be added.
-
line 108. It is reported the Tl dopant levels in the initial powder; it should be added the Tl concentrations for the grown crystal, if available
-
Fig 2, it is better to invert the two plots, following the logics of the analysis done: before the selection of neutrons in the BD, afterwards the selection of the TOF.
-
line 159. At the end of the sentence remove the colon

Author: Mukund Raghunath Bharadwaj on 2022-11-03 [id 2979]
(in reply to Report 1 on 2022-10-12)Thank you for the suggestions in this regard. I implemented all the necessary changes aforementioned and prepared an updated manuscript.
Regarding the Tl dopant concentrations reported in line 108, the Tl concentrations of the tested crystals are still yet to be verified and hence was not mentioned yet. (although the location in the crystal boules that they were cut from was reported to have almost the same Tl dopants as the initial powder from SICCAS)
Anonymous on 2022-11-08 [id 2995]
(in reply to Mukund Raghunath Bharadwaj on 2022-11-03 [id 2979])Please resubmit an updated version.

---

## Round 3 · Referee Report · Anonymous (Referee 1) · 2022-12-12

Report

The paper can be published now.

---

## Editorial Decision

published